# Chain mediating effect of self-esteem and resilience on the relationship between wellbeing literacy and subjective well-being among college students

Ying Xie[1], Kai Zeng[2], Yun Yang[3], Fangfang Zheng[4]*, Feifei Wang[5]*

**1** School of Modern Business and Trade, Chongqing Business Vocational College, Chongqing, China,
**2** School of Foreign Languages and Literature, Chongqing Normal University, Chongqing, China,
**3** College of Physics and Electronic Engineering, Chongqing Normal University, Chongqing, China,
**4** School of Basic Medicine, Army Medical University, Chongqing, China, **5** School of Medical Psychology, Army Medical University, Chongqing, China

* dudu_fiona@163.com (FZ); wff_0918@163.com (FW)

## Abstract

Based on the framework of psychological capital theory from the perspective of positive psychology, this study explores the influence mechanism of wellbeing literacy on the subjective well-being of college students. In recent years, with the increasing social attention to psychological well-being, wellbeing literacy, as the core element of individuals' ability to achieve -well-being, has gradually become a focal point in positive psychology research. As a new model for well-being science and practice, wellbeing literacy can provide a tangible way to evaluate mechanisms learned from well-being interventions. This study attempts to investigate both the influence of wellbeing literacy on college students' subjective well-being and the mediating role of self-esteem and resilience in this relationship. A total of 1030 college students were surveyed using the wellbeing literacy, self-esteem, simplified resilience, life satisfaction, and positive and negative emotion scales. The study results reveal that (1) Wellbeing literacy significantly impacts subjective well-being both overall and directly; (2) Self-esteem significantly mediates the relationship between wellbeing literacy and subjective well-being; (3) A significant chain-mediating effect exists through self-esteem and resilience; (4) The chain-mediating role explains the mechanism underlying this relationship. These findings confirm that wellbeing literacy not only directly promotes subjective well-being, but also cascades through self-esteem enhancement and subsequent resilience improvement. This study provides empirical support for current well-being theory, and recommends stepwise interventions focusing on well-being language cultivation: consolidating the foundation of self-esteem through cognitive reconstruction in early stages, and strengthening stress response training to improve resilience in later phases. Future research can further verify the timing effects of pathways in combination with a tracking design. As a new model

**Data availability statement:** The research data for this study has been uploaded to the Zenodo platform. Data are available at https://zenodo.org/records/15350904.

**Funding:** This research was funded by the ninth batch of pilot projects of comprehensive education reform in Chongqing, fund number: 23JGS63. The Humanities and Social Science Research Project of Chongqing Education Commission, fund number: 2023SJYB0130. Chongqing Higher Education Examination Recruitment Research Project for the Year 2024, fund number: CQZSKS2024021. Chongqing Education Science Planning Project General Project: Research on Dynamic Identification and Intervention Strategies of College Students' Psychological Crisis Based on Fisher Discriminant Method, fund number: K24YG3260177. The National Social Science Fund Project of China, fund number: 19XSH018. The funds had no role in study design, data collection and analysis, decision to publish, or preparation of the manuscript.

**Competing interests:** The authors declare that the research was conducted in the absence of any commercial or financial relationships that could be construed as a potential conflict of interest.

for well-being science and practice, wellbeing literacy can provide a tangible way to evaluate mechanisms learned from well-being interventions.

## 1 Introduction

Promoting the mental health of students has become an important issue in higher education around the world. Growing attention has been paid to cultivating individuals' well-being abilities through positive education interventions, which encompass not just the cultivation of positive emotions, cognition and behaviors [1]. This type of wellbeing literacy is a positive psychological intervention from a developmental perspective [2]. Subjective well-being mainly refers to individuals' feelings and evaluations of their life satisfaction, while wellbeing literacy encompasses the understanding and pursuit of well-being through cognitive, emotional and behavioral dimensions. As an emerging concept, wellbeing literacy emphasizes constructing a cognitive framework of well-being through language symbolic systems, and its correlation mechanisms with subjective well-being need to be discussed in depth. Focusing on the college student population, this study aims to reveal potential pathways for improving subjective well-being through wellbeing literacy, with particular attention to the mediating mechanisms of self-esteem and resilience, to provide theoretically based intervention solutions for mental health education in colleges and universities.

Wellbeing literacy refers to the skills used to consciously maintain or improve an individual's or others' vocabulary, knowledge and language about well-being, focusing on using language in life to promote well-being, with this concept being discussed by Oades et al. [2]. Therefore, it helps individuals and others get well-being if they have more ability to apply language [3,4]. As shown in Fig 1, the content of wellbeing literacy comprises five components as identified by Oades et al. [2]: vocabulary and knowledge of well-being, understanding multimodal texts related to well-being [5], composing multimodal texts related to well-being, context awareness and adaptability, and intentionality for well-being [6]. Wellbeing literacy can help improve well-being experiences by serving as an intermediary role, which means that the more choices individuals have to "transform" well-being opportunities into significant well-being experiences in their internal or external environment at higher levels of wellbeing literacy [7].

Subjective well-being encompasses people's evaluation of their lives, including cognitive evaluation (such as life satisfaction) and emotional experiences (such as positive or negative emotions) [8]. Research suggests that individuals with higher levels of well-being tend to exhibit more satisfactory behaviors than those with lower levels in various life domains. According to Lester et al. [9], it is evident that military performance can be improved by enhancing well-being. John et al. [10] suggest that subjective well-being has a positive impact on health, longevity, supportive social relationships, citizenship, job performance, and resilience.

Nevertheless, there is a paucity of research that has further explored the relationship between self-esteem, resilience, wellbeing literacy and subjective well-being. This research seeks to utilize the concept of wellbeing literacy as a conduit, enabling

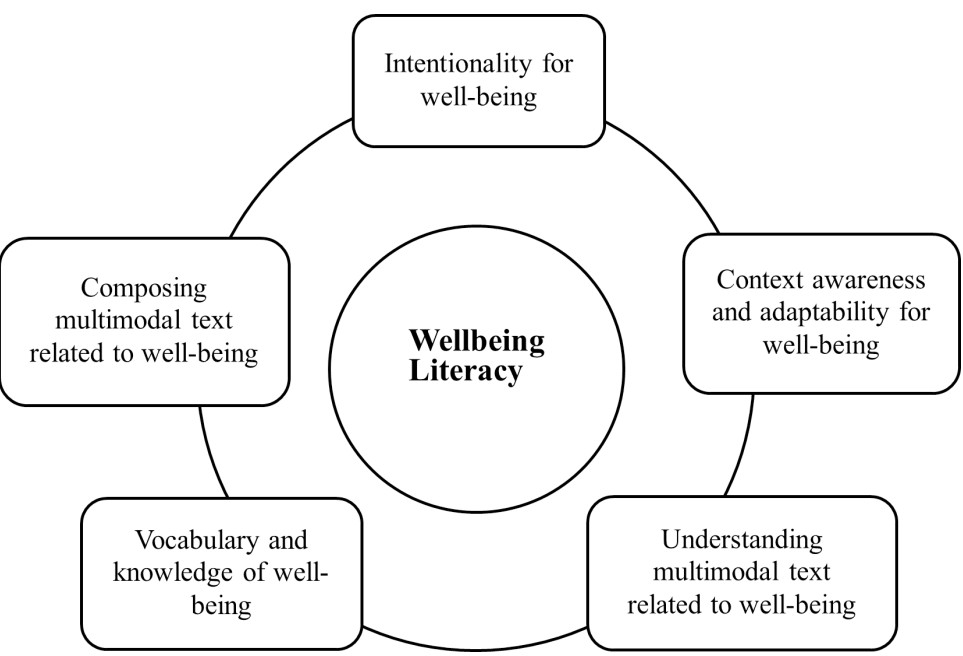

**Fig 1. Five Components of Wellbeing Literacy.**

individuals to proactively identify and transform positive cues in their environment through the language symbolic system (e.g., reconceptualising academic pressure as growth opportunities) to establish a primary cognitive reserve. Individuals then combine resources inherent within the self-system (self-esteem) to filter negative information through an active self-schema, thereby maintaining a stable sense of self-worth and self-efficacy. Concurrently, they utilize resilience as a focal point to minimize resource loss through adaptive coping strategies in high-pressure situations, thereby ensuring the sustainability of their well-being experiences. The central tenet of this process is the dynamic accumulation and optimal allocation of psychological resources, which are integral to enhancing well-being.

This study establishes a novel chain model of "well-being cognitive reconstruction – self-positive resource gain – psychological adaptation strengthening – well-being enhancement" within the Asian cultural context for the first time, marking a hitherto unexplored area. The proposed model systematically elucidates the sequential mechanisms of positive psychological resources during well-being competence transformation.

## 1.1 The relationship between wellbeing literacy and subjective well-being

The primary distinction between wellbeing literacy and subjective well-being lies in the former's reliance on the ability to utilize and recognize well-being language to establish lasting well-being [11]. Oades et al. [12] presented their recent findings at the Fifth International Conference on Positive Psychology in China. The authors propose that wellbeing literacy mediates the relationship between the environment—both internal and external—and well-being experiences, positing that higher literacy levels correlate with a heightened sense and prolonged duration of well-being within the same settings. Despite the absence of specific findings regarding the direct relationship between wellbeing literacy and subjective well-being, mental well-being exerts a significant influence on mental health, shaping its quality and fostering a positive interplay between the two [13]. It is posited that, given the capacity of wellbeing literacy to engender a sense of well-being experience, subjective evaluations of well-being may also exert a positive effect. Based on the above analysis, we propose the following hypothesis:

**Hypothesis 1:** Wellbeing literacy positively predicts college students' subjective well-being.

## 1.2 The relationship between self-esteem, wellbeing literacy and subjective well-being

Self-esteem refers to the evaluation of one's own worth from subjective perspective [14,15]. In the review by Campbell et al. [16], individuals with higher levels of self-esteem tend to view themselves positively. In the face of failure, they not only adopt positive coping styles but also filter out negative information, resulting in improved subjective well-being [17]. In a cross-cultural study of 13,118 college students from 31 countries, Diener & Diener [18] found a significant positive correlation between self-esteem and life satisfaction component of subjective well-being. The wellbeing literacy ability model suggests that internal personality traits enhance wellbeing literacy abilities [12]. It encompasses the concept of "multidimensional meaning of well-being intentions [7], which facilitates the active identification of positive cues in the environment (e.g., opportunities for success, social support). This process can be conceptualized as an accumulation of cognitive resources. Consequently, self-esteem, as a positive resource for self-evaluation, may encourage individuals to explore positive cognition, actively seize opportunities to practice and transform them into well-being achievements, and shift from static subjective well-being experiences to dynamic wellbeing literacy skills. The preceding analysis leads to a secondary hypothesis.

**Hypothesis 2:** Self-esteem plays a mediating role in the relationship between wellbeing literacy and subjective well-being.

## 1.3 The relationship between resilience, wellbeing literacy and subjective well-being

Resilience is the capacity to bounce back from setbacks, learn from failures, draw motivation from challenges, and retain the belief that one can overcome stressors or difficulties in life [19]. There is a positive correlation between resilience and subjective well-being, suggesting that stronger resilience corresponds to greater subjective well-being [20]. According to Ma et al [21], resilience emerges from the interaction between internal and external factors, as demonstrated in the theory of resilience. The theory of wellbeing literacy emphasizes that individuals should actively select and transform opportunities to achieve well-being in both their internal and external environments, utilizing the positive role of resilience in coping strategies. This argument is supported by research from Jia et al. [11], which suggests that military cadets inevitably encounter adverse experiences caused by stressful events such as stress and setbacks, and the wellbeing literacy of military cadets can mitigate these bad experiences by influencing subjective well-being [11] through resilience. Resilience, as an important protective resource, has been examined as a mediating variable in many studies. For example, Murat and Gokmen [22] found that resilience plays a mediating role in the relationship between hope, mental health, and subjective well-being. It can thus be concluded that individuals have the capacity to actively reassess stressful events by applying well-being language skills to reduce threat perception. Concurrently, enhanced resilience fosters positive well-being experiences by drawing on learned strategies, ultimately improving subjective well-being. This paper puts forward the third hypothesis.

**Hypothesis** 3: Psychological resilience plays a mediating role in the relationship between wellbeing literacy and subjective well-being.

## 1.4 The relationship between self-esteem, resilience, wellbeing literacy and subjective well-being

Several studies have proved that self-esteem can act as an intermediary factor in facing adversity [23–25], which can have a positive impact on coping with stressors and may improve overall well-being. Furthermore, individuals are less susceptible to behavioural and emotional issues when they possess high levels of self-esteem and resilience [23]. These problems include depression, anxiety [26], and loneliness [27]. Other studies [28,29] have also found that young people who are more resilient tend to have greater self-esteem and are less likely to engage in risky behaviors than their less

resilient peers. Resilience alleviates mental health issues and well-being in adversity [23,30] and focuses on its mediating role [31,32]. Some current studies have used self-esteem and resilience as mediating variables. Ma found that left-behind teenagers' understanding of the influence of social support on subjective well-being occurs mainly through self-esteem, and resilience plays a chain intermediary role [21]. In his 2019 study, Gökmen [33] researched the correlation between social exclusion and life satisfaction. The study found that resilience and self-esteem play a full mediating role in this relationship. Acting as one of the indicators of subjective well-being, life satisfaction is important. It can be hypothesised that if wellbeing literacy encourages individuals to have cognitive reserves for well-being by applying language, self-esteem may help individuals buffer stress, promote the generation of resilience, and thus maintain emotional stability and stimulate subjective well-being. The accumulation of positive psychological resources may follow the sequential strengthening path of "cognitive storage → self-worth identification → stress adaptation → subjective well-being enhancement", providing a theoretical basis for subsequent systematic psychological intervention. The present study puts forward the fourth hypothesis.

**Hypothesis 4:** Self-esteem and resilience play a chain mediating role in the relationship between wellbeing literacy and subjective well-being.

The present study is founded upon two theories: the resource preservation theory [34] and the psychological capital model [35]. The study hypothesises that wellbeing literacy can function as an individual's cognitive resource reserve, aiding in the identification and transformation of positive environmental cues (e.g., the attainment of happiness through the utilization of language skills). Self-esteem is defined as a self-system resource that protects individuals from the erosion of negative information and maintains positive self-evaluation, while resilience is used as an adaptive resource to reduce resource loss in stressful situations and promote subjective well-being. The present study proposes an integrated framework: wellbeing literacy as cognitive reserve can ultimately enhance subjective well-being, with gains in self-esteem levels (self-system resources) and resilience (adaptation system resources).

## 2 Materials and methods

### 2.1 Participants and procedure

This cross-sectional study recruited 1030 college students using convenience sampling between October 8, 2022, and October 26, 2022. The sample size of 1030 was determined via convenience sampling and was deemed sufficient to provide over 80% power to detect small to medium effects (e.g., correlations of 0.1 or greater) in the planned correlation, regression, and mediation analyses, consistent with statistical guidelines and practices in similar studies [36].

A summary of the demographic statistics is presented: 353 (34.3%) participants were male, while 677 (65.7%) were female. The sample consisted of 430 freshmen (41.8%), 323 sophomores (31.4%), and 276 juniors (26.8%). Additionally, 237 participants (23%) were only children, 493 participants (47.9%) had been left behind for over half a year before the age of 10, and 185 (18%) were from single-parent or divorced families. The average age was $19.14 \pm 1.12$ years (95% CI: 19.07–19.21).

Psychology teachers as the main test use of the questionnaire star platform (https://www.wjx.cn/) in batches of collective measurement to measure 50–100 college students at a time in this study. Even after voluntary participation, they were allowed to withdraw from the evaluation at any time without providing any reason. They were also assured that personal information would not be disclosed. After the evaluation, the data are sorted out and coded.

### 2.2 Ethical approval

The study has been conducted in compliance with the Helsinki Declaration and approved by the Ethics Committee of the Army Military Medical University. All participants signed an electronic informed consent form prior to the start of the study.

## 2.2 Measures

**2.2.1 Wellbeing literacy.** The Chinese version of *the Wellbeing Literacy 6 Item Scale (Well-Lit 6)* is based on the definition of literacy provided by the *Australian Curriculum, Assessment and Reporting Authority (ACARA)*. The scale was scored by Likert 7 ranging from 1 (strongly disagree) to 7 (strongly agree. The internal consistency coefficient was higher in these college students (students: Chisel = 0.84, faculty: Chisel = 0.91, parents: Chisel = 0.91) [7,11]. The Cronbach's coefficient was 0.98.

**2.2.2 Self-esteem.** The Rosenberg Self-Esteem Scale (SES) was utilized to measure the overall self-esteem of college students [37]. The scale consists of 10 items rated on a 4-point Likert scale (1 = strongly disagree; 4 = strongly agree), with 5 reverse-scored (items 3, 5, 8, 9, 10). According to Shen and Cai [38], cultural differences between China and the West necessitate either positive scoring or deletion of Item 8 ("I hope I can win more respect for myself"). This study scores this item positively. Higher total scores indicate higher self-esteem. The internal consistency coefficient was 0.895.

**2.2.3 Resilience.** The Brief Resilience Scale (BRS), compiled by Smith et al. [39] consists of 6 items (3 positively scored, 3 negatively scored). Responses are recorded on a Likert 5-point scale (1 = strongly disagree; 5 = strongly agree). This study uses Chen et al.'s [40] revised Chinese version, designed to measure individuals' ability to regain well-being when coping with stress. The internal consistency coefficient was 0.794.

**2.2.4 Subjective well-being.** Based on Deci and Ryan's framework, subjective well-being comprises both cognitive and emotional components [41]. Cognitive well-being was measured by Satisfaction with Life Scale (SWLS) developed by Diener et al. [42], with the Chinese version translated and revised by Wang et al. [43]. The 5-item scale (e.g., "Most of what I live is close to what I'd like to live") uses a 7-point Likert scale (1 = strongly inconsistent, 7 = strongly consistent), where higher scores indicate higher cognitive well-being ($\alpha = 0.892$). Emotional well-being was assessed via the emotional scale compiled by Watson [44].

## 2.3 Data analysis

**2.3.1 Common method bias test.** Given the self-reported nature of the data, anonymous processing and reverse scoring of some items were employed to control bias. Harman's single-factor test [45] revealed eight factors with eigenvalues >1. The first factor explained 37.902% of variance, which is slightly below the critical threshold of 40%, indicating no significant common method bias.

**2.3.2 Analysis method.** SPSS 24.0 was used for descriptive statistical analysis, correlation analysis, and regression analysis ($\alpha = 0.05$). In addition, Pearson correlation examined relationship among wellbeing literacy, self-esteem, resilience and subjective well-being. To assess the significance of the intermediary effect model, the bootstrap resampling program 5000 times is utilized. In this study, we employed bootstrapping to enhance the reliability of our Confirmatory Factor Analysis (CFA) results, particularly given the sample size constraints. Bootstrapping is a resampling technique that generates multiple simulated samples from the original dataset, allowing for the estimation of the sampling distribution of a statistic. This approach offers several advantages, especially in the context of CFA. First, bootstrapping provides more accurate standard errors and confidence intervals for model parameters, which is particularly beneficial when the sample size is limited or when the data may not fully meet normality assumptions [46] Second, it helps assess the stability and robustness of the model estimates by reducing the impact of sampling variability.

It is considered the effect significant when the 95% confidence interval does not include 0. The double-tailed test is used to calculate the P value. In order to verify the model fitting, this paper used the following fitting indexes: $\chi^2$ test (P < 0.05), $\chi^2/df$ < 3, comparison fitting index (CFI > 0.90) incremental fitting (IFI > 0.90), goodness of fit index (GFI > 0.90), Tucker-Lewis index (TLI > 0.90), approximate root mean square error (RMSEA <0.05) [47].This study adhered to the STROBE cross-sectional reporting guidelines, ensuring clear and transparent reporting of all methodological aspects.

## 3 Results

### 3.1 Descriptive statistics and correlation analysis

Descriptive statistics and correlations are presented in Table 1, displaying the average, standard deviation, 95% confidence interval, and correlation matrix for each variable. Pearson analysis revealed significant positive correlations between wellbeing literacy and self-esteem, resilience and subjective well-being, supporting subsequent mediation analysis.

### 3.2 Relationships between variables

The SEM analysis results (Table 1) indicate that wellbeing literacy has a significant positive impact on self-esteem (β = 0.757, SE = 0.019, p = 0.001) and subjective well-being (β = 0.206, SE = 0.045, p < 0.001), but not on resilience (β = 0.109, SE = 0.060, p = 0.072). Furthermore, it is worth noting that self-esteem significantly predicts resilience (β = 0.619, SE = 0.058, p = 0.001) and subjective well-being (β = 0.491, SE = 0.052, p = 0.001). Resilience also positively affects subjective well-being (β = 0.350, SE = 0.041, p < 0.001).

### 3.3 Regression analysis of wellbeing literacy, self-esteem and resilience on subjective well-being

Stepwise regression analysis was conducted with subjective well-being (Y) as dependent variable and wellbeing literacy, self-esteem and resilience as independent variables. Wellbeing literacy (X1), self-esteem (X2) and resilience (X3) as independent variables, the result shows that Variance Inflation Factor (VIF) of the predicted variables are all less than 5 indicating that there was no multicollinearity problem in the regression equation. The regression equation $Y = -12.137 + 0.092X_1 + 0.190X_2 + 0.177X_3$ is obtained. Self-esteem, wellbeing literacy and resilience are positively correlated with subjective well-being. The decision coefficient of the regression model is $R^2 = 0.642$, and the adjusted decision coefficient is $R^2 = 0.641$, which can explain the 64.1% variation of subjective well-being as shown in Table 2 below.

### 3.4 Mediation effect test

Based on correlation analysis and regression model, the structural equation model (SEM) was utilized to analyze the relationship among wellbeing literacy, self-esteem, resilience with subjective well-being. The good model fit indices suggest that the hypothetical model is valid (χ2/df = 1.880, RMSEA = 0.029, GFI = 0.975, AGFI = 0.957, CFI = 0.963, TLI = 0.947).

**Table 1. Descriptive Statistics and Correlation analysis (*r*).**

| Variables | $\bar{x} \pm s$ | 95%IC | 1 | 2 | 3 | 4 |
|---|---|---|---|---|---|---|
| 1 Wellbeing literacy | 31.08 ± 6.54 | 30.68,31.48 | 1 | | | |
| 2 Self-esteem | 29.86 ± 4.68 | 29.57,30.14 | .683** | 1 | | |
| 3 Psychological Resilience | 20.30 ± 3.81 | 20.07,20.53 | .480** | .588** | 1 | |
| 4 Subjective well-being | 0.00 ± 2.32 | −0.14,0.14 | .663** | .734** | .642** | 1 |

Note.** *p* < 0.01.

**Table 2. Stepwise Regression Analysis of Self-esteem, Wellbeing Literacy and Resilience on Subjective Well-being.**

| Variables | Non-standardized regression coefficient | Standard error | Standardized regression coefficient | t | *P* |
|---|---|---|---|---|---|
| Constant | −12.137 | 0.292 | | −41.571 | <0.001 |
| Wellbeing Literacy | 0.092 | 0.009 | 0.260 | 13.757 | <0.001 |
| Self-esteem | 0.190 | 0.014 | 0.385 | 12.511 | <0.001 |
| Resilience | 0.177 | 0.014 | 0.292 | 10.075 | <0.001 |

The confidence interval method was used to test indirect effects using Amos 24.0 software. The results show that the total effect of the mediation model is 0.517 (95%CI: 0.464–0.568, $p = 0.001$). This finding supports Hypothesis 1, indicating a positive correlation between wellbeing literacy and subjective well-being. The direct effect is 0.137 (95% CI: 0.075–0.195, $p < 0.001$), accounting for 26.50% of the total effect. Meanwhile, the indirect effect size is 0.247 (95% CI: 0.192–0.305, $p < 0.001$), accounting for 47.78% of the total effect, which proves that wellbeing literacy significantly mediates subjective well-being through self-esteem, supporting Hypothesis 2. However, the indirect effect is 0.025 (95% CI: −0.001–0.056, $p = 0.064$), indicating that the mediating effect of wellbeing literacy on subjective well-being through resilience is not significant, which contradicts Hypothesis 3.

In addition, self-esteem and resilience has been revealed to play a notable mediating role between wellbeing literacy and subjective well-being, accounting for 21.08% of the total effect (indirect effect: 0.109, 95% CI: 0.079–0.144, $p < 0.001$). Hypothesis 4 is supported. The mediation model is shown in Fig 2, and effect analyses are summarized in Table 3.

## 4 Discussion

The study revealed a direct correlation between wellbeing literacy and subjective well-being, self-esteem, and resilience. Although previous research has explored these relationships, few studies have systematically explained the mechanisms. Three of the four hypotheses were supported, with Hypothesis 3 (resilience as an independent mediator) being the exception.

First, the study demonstrated a significant positive correlation between wellbeing literacy and subjective well-being, aligning with Hou et al.'s [7] findings and supporting the wellbeing literacy ability model, indicating that college students who actively seek well-being opportunities enhance their experiences and develop well-being competencies. Cultivating these abilities is crucial for long-term mental health maintenance. Notably, this study bridges a key research gap by establishing the first empirical link between wellbeing literacy and self-esteem. Self-esteem, defined as an individual's sense of value and capabilities [48], may be strengthened through wellbeing literacy, as individuals with higher literacy use

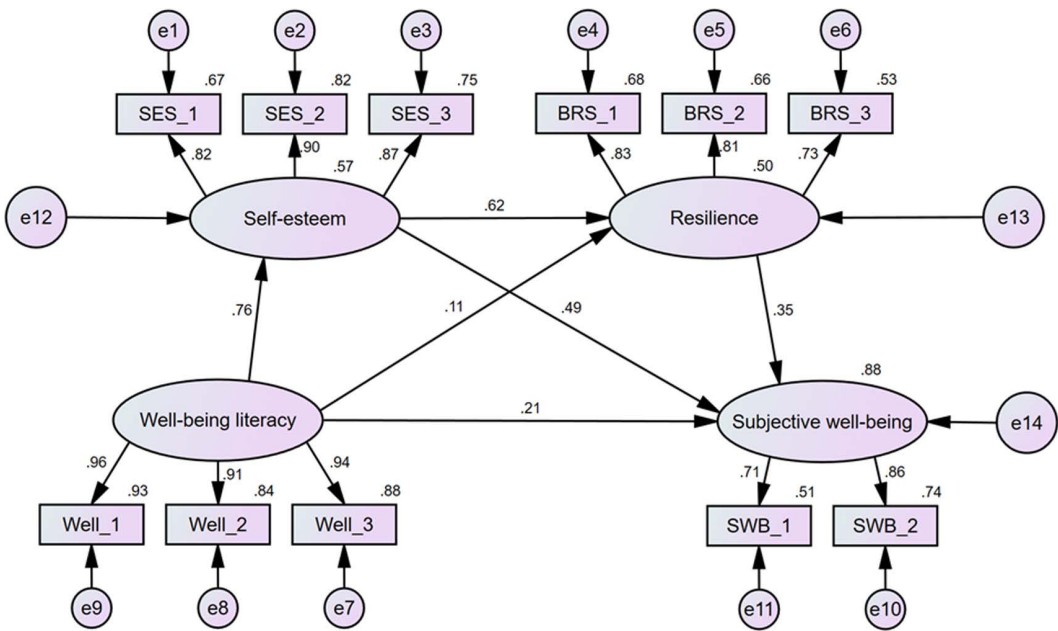

**Fig 2. The Relationship Between Wellbeing Literacy, Self-Esteem, Resilience, and Subjective Well-Being.**

**Table 3. Analysis of Mediating Effects.**

| Relationships | Point Estimate | Product of Coefficients | | Bootstrapping 5000 times 95% confidence interval | | | | | |
|---|---|---|---|---|---|---|---|---|---|
| | | | | Bias-Corrected 95% CI | | | Percentile 95% CI | | |
| | | SE | Z | Lower | Upper | P | Lower | Upper | P |
| **Direct effects** | | | | | | | | | |
| Wellbeing Literacy-SES | 0.352 | 0.015 | 23.467 | 0.32 | 0.381 | 0.001 | 0.323 | 0.385 | 0.000 |
| Wellbeing Literacy-BRS | 0.061 | 0.034 | 1.794 | −0.005 | 0.128 | 0.072 | −0.006 | 0.128 | 0.073 |
| Wellbeing Literacy-SWB | 0.137 | 0.03 | 4.567 | 0.075 | 0.195 | 0.000 | 0.076 | 0.196 | 0.000 |
| SES-BRS | 0.75 | 0.074 | 10.135 | 0.592 | 0.891 | 0.001 | 0.604 | 0.898 | 0.000 |
| SES-SWB | 0.701 | 0.076 | 9.224 | 0.556 | 0.853 | 0.000 | 0.559 | 0.855 | 0.000 |
| BRS-SWB | 0.413 | 0.051 | 8.098 | 0.315 | 0.517 | 0.000 | 0.312 | 0.513 | 0.000 |
| **Indirect effects** | | | | | | | | | |
| Wellbeing Literacy to SES to SWB | 0.247 | 0.029 | 8.517 | 0.192 | 0.305 | 0.000 | 0.194 | 0.307 | 0.000 |
| Wellbeing Literacy to BRS to SWB | 0.025 | 0.015 | 1.667 | −0.001 | 0.056 | 0.064 | −0.002 | 0.055 | 0.073 |
| Wellbeing Literacy to SES to BRS to SWB | 0.109 | 0.017 | 6.412 | 0.079 | 0.144 | 0.000 | 0.078 | 0.143 | 0.000 |
| **Total indirect effects** | 0.381 | 0.027 | 14.111 | 0.328 | 0.434 | 0.001 | 0.331 | 0.438 | 0.000 |
| **Total effects** | 0.517 | 0.026 | 19.885 | 0.464 | 0.568 | 0.001 | 0.468 | 0.572 | 0.000 |

language skills to positively reframe self-worth and articulate well-being traits. This process parallels how cultural environments shape well-being perceptions, akin to natural environmental influences.

Second, our findings extend prior work on wellbeing literacy and resilience. While Jia et al. [49] reported a significant positive correlation between wellbeing literacy and resilience, this aligns with our observations. Educating and guiding college students to understand and articulate well-being through language is a positive practice, as it not only fosters meaningful well-being achievements but also strengthens resilience against external setbacks [50]. Additionally, previous studies [51,52] highlight consistent positive correlations among subjective well-being, self-esteem, and resilience. Thus, enhancing self-esteem and resilience serves as a measure for fostering college students' well-being.

The stepwise regression analysis shows that wellbeing literacy positively affects subjective well-being, aligning with recent studies [49]. This study employs structural equation modeling to examine the mediating roles of self-esteem and resilience between wellbeing literacy (the independent variable) and subjective well-being (the dependent variable). The findings reveal that self-esteem acts as an independent mediator, while resilience does not significantly mediate, indicating that wellbeing literacy influences subjective well-being via self-esteem rather than resilience. This is because resilience is typically activated in the context of a risk environment characterized by difficulties, tensions, or threats [53,54]. Nevertheless, wellbeing literacy, as a positive psychological resource, does not activate resilience to influence college students' subjective well-being. This finding contrasts with the results reported by Jia et al. [49]. In China, college students are not subjected to military-style management, nor are they concurrently burdened with training and heavy academic workloads. Conversely, after the college entrance examination, students are less likely to encounter a risk environment, making resilience less likely to be activated. Additionally, the study demonstrates that self-esteem and resilience significantly mediate the relationship between wellbeing literacy and subjective well-being when included together in the model. In other words, wellbeing literacy can indirectly influence subjective well-being levels via self-esteem and resilience. This study's findings contribute to the theoretical research on wellbeing literacy and subjective well-being by addressing a gap in the literature. Oades et al. [12] define wellbeing literacy as the ability to use language to exert control over one's environment, make beneficial choices, and achieve well-being experiences, thereby enabling college students with high wellbeing literacy to perceive a more meaningful well-being experience through their language knowledge and application skills. This also supports the notion that individuals who actively select positive coping strategies experience a heightened sense of well-being

[55,56]. Students with high wellbeing literacy are inclined to engage in beneficial activities, foster high self-esteem, and accumulate positive resources for adversity, leading to enhanced well-being achievement experiences.

## 5 Limitations

The use of self-report scales in this study may introduce social desirability bias, potentially compromising the ecological validity of the findings. To address these limitations, future research could employ mixed-method approaches, such as empirical behavioral sampling and physiological indicator monitoring to triangulate data sources. Additionally, a multi-dimensional well-being assessment system can be constructed by integrating other evaluation methods, such as situational experiments (e.g., resilience under simulated stressors) and neuroimaging techniques (e.g., prefrontal cortex activation patterns), with the aim of mitigating measurement biases and enhance robustness.

While this study focused on internal positive factors (e.g., self-esteem and resilience) influencing wellbeing literacy, future investigations should incorporate cognitive variables (e.g., self-efficacy, growth mindset) and external protective factors (e.g., mentor support and family cohesion) to construct a hybrid "individual-relationship-environment" model. Such a framework. would clarify boundary conditions governing wellbeing literacy mechanisms.

The present study is limited to Chinese college students; however, cross-cultural comparative research could be constructed in future stages to verify the model's robustness. Concurrently, future research should also explore practical applications, such as:

1. Implementing a dual-model intervention strategy that integrates "wellbeing literacy" and "stress exposure" to determine resilience thresholds;

2. Investigating the interactive effects of self-esteem and resilience on wellbeing literacy and subjective well-being during critical transitions (e.g., employment);

3. Incorporating wellbeing literacy into university mental health curricula to establish cross-institutional monitoring frameworks for the "wellbeing literacy-self-esteem- resilience-subjective well-being" pathway.

## 6 Conclusion

Due to the current economic downturn and other challenges, college students are facing increasing psychological pressure. It is crucial to address these issues proactively and provide support for students striving to complete their studies, develop professional competencies, and pursue fulfilling lives. However, prolonged exposure to such stressors may exacerbate negative emotions and diminish well-being levels. This research offers a novel perspective on enhancing college students' well-being, particularly through the cultivation of wellbeing literacy as a practical skill. When students possess a positive internal environment (e.g., high self-esteem), their wellbeing literacy (knowledge, skills, etc.) is enhanced [12], enabling proactive engagement with external environment and thereby improving subjective well-being. Self-esteem and resilience, as positive psychological traits, serve as sequential mediators in the relationship between wellbeing literacy and subjective well-being. Moreover, wellbeing literacy exerts a direct and significant impact on subjective well-being.

The findings provide both a theoretical foundation and practical guidance for designing active interventions in higher education. We recommend that institutions:

1. Implement primary prevention by integrating wellbeing literacy courses to strengthen cognitive reserves;

2. Prioritize secondary reinforcement through self-esteem enhancement techniques (e.g., achievement event reflection) to build psychological resource buffers.

3. Adopt tertiary integration by embedding resilience training (e.g., frustration simulation) in high-stress contexts like career planning, fostering synergistic improvements in "well-being, cognition, positive emotions, positive behaviour".

Future research should validate the temporal dynamics of these pathways using longitudinal designs. This study proposes a chain-mediated model—"cognitive reconstruction → resource gain → adaptation strengthening → well-being enhancement"—that systematically elucidates the sequential mechanisms through which positive psychological resources operate within wellbeing literacy frameworks. We advocate for policy initiatives to prioritize the systematic integration of wellbeing literacy into educational curricula.

## Author contributions

**Conceptualization:** Ying Xie, Feifei Wang.

**Data curation:** Kai Zeng, Fangfang Zheng.

**Formal analysis:** Ying Xie, Feifei Wang.

**Funding acquisition:** Ying Xie, Yun Yang, Feifei Wang.

**Methodology:** Ying Xie, Feifei Wang.

**Writing – original draft:** Ying Xie.

**Writing – review & editing:** Feifei Wang.

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
