## [Decision Letter · Decision Letter 0]

12 Mar 2025

Dear Dr. Feifei Wang,

Thank you for submitting your manuscript to PLOS ONE. After careful consideration, we feel that it has merit but does not fully meet PLOS ONE’s publication criteria as it currently stands. Therefore, we invite you to submit a revised version of the manuscript that addresses the points raised during the review process.

We look forward to receiving your revised manuscript.

Kind regards,

Maria José Nogueira, Ph.D.

Academic Editor

PLOS ONE

Journal Requirements:

“Thank you for stating the following in your Competing Interests section: “

Please complete your Competing Interests on the online submission form to state any Competing Interests.

4. In the online submission form, you indicated that [Insert text from online submission form here].

Additional Editor Comments:

Dear author

See reviewer recommendations.

Best regards

Reviewers' comments:

Reviewer's Responses to Questions

**Comments to the Author**

1. Is the manuscript technically sound, and do the data support the conclusions?

Reviewer #1: Partly

2. Has the statistical analysis been performed appropriately and rigorously?

Reviewer #1: Yes

3. Have the authors made all data underlying the findings in their manuscript fully available?

Reviewer #1: Yes

4. Is the manuscript presented in an intelligible fashion and written in standard English?

Reviewer #1: No

Reviewer #1: I appreciate the opportunity to review this manuscript and hope my comments assist in the revision process. The material is interesting and the topic is timely and relevant. The method seems to have been followed faithfully and the authors were well-positioned to conduct the analysis. Despite these positives, in my view, the paper needs more work before it could be published and I have made some specific suggestions below.

- The abstract in question would benefit from some form of framing of the context of the study, by which I mean the history of the problem and the results already formulated that are relevant to it. This would allow a better understanding of the importance of the topic. Besides, improve your conclusions.

- Keywords: are these keywords Mesh terms?

- The literature addressed is described accurately so far as I can see. However, there is no clear distinction between manuscript sections in terms of the content they report. First, I suggest dividing the section "INTRODUCTION" into three components, respectively introduction (explain the general argument of the paper, without going into specific details) background (situate the study concepts within the context of extant knowledge, discuss the international relevance of the concepts) and purpose, creating greater clarity in the analysis of the reader. What is the study's biggest contribution? The contribution should be clearly stated in the introduction.

- Ensure that the hypotheses follow logically from the literature review presented. Each hypothesis should build on the arguments presented in the introduction.

- Since there are multiple concepts and components that are being discussed in the Introduction section, the authors are encouraged to create a framework or diagram to show the relationships between the different factors.

Method

- Was this study part of a larger study with more variables? Has any of this data been published?

- In your methods section, say that you used the STROBE cross-sectional reporting guidelines. Is it necessary to clarify the method employed?

- Were the sample sizes sufficiently powered to detect effects? How did the researchers decide on sample size?

- Was there compensation for participating in the study?

- Provide details on the adequacy of the sample size for CFA.

- Discuss the advantages of bootstrapping in more detail.

- Explain how the covariates were selected and included in the mediation analysis. Discuss the potential impact of these covariates on your results.

- The process of analysis should be made as transparent as possible. What strategies were used to avoid duplications or fraud in the online survey? Did you analyze any potential non-response bias? And early vs late bias? Did you check if data can suffer from common method bias?

Results

- A better visual structure of tables (boldface variables with statistical significance) would improve the readability.

- How authors handled missing data?

Discussion

- Some of the contributions that are highlighted here could be flagged in the introduction for a more consistent narrative throughout the paper. I believe there should be better integration of the results with the existing literature.

- A stronger discussion of implications for future research and potential intervention work is needed. Identify recommendations for practice/research/education/management as appropriate, and consistent with limitations, in order to more fully allow readers to understand the extent to which the authors were able to answer the research questions and to grasp the limitations of this study.

- Theoretical and methodological limitations should be emphasized more deeply.

CHECKLIST FOR STYLE

The manuscript will serve a broad audience of students, researchers, and practitioners, however, the manuscript needs to be carefully and attentively proofread, because some sentences are awkwardly constructed, punctuation is deficient, and therefore reading is occasionally difficult to follow. That leads me to believe that it needs careful editing by a native English speaker.

**Do you want your identity to be public for this peer review?** For information about this choice, including consent withdrawal, please see our Privacy Policy

Reviewer #1: No

---

## [Author Response · Author response to Decision Letter 1]

8 May 2025

Dear reviewers,

Thank you for providing the opportunity to revise our manuscript and for the constructive feedback from the reviewers.

1.The abstract in question would benefit from some form of framing of the context of the study, by which I mean the history of the problem and the results already formulated that are relevant to it. This would allow a better understanding of the importance of the topic. Besides, improve your conclusions.

Response: I added the background of this study at the beginning of the abstract, as detailed in lines 17-21, and optimized the conclusions from practical significance and future interventions, as detailed in lines 33-39.

2.Keywords: are these keywords Mesh terms?

Response: Thank you for the reviewer's attention to whether our keywords are MeSH terms. After investigation, we found that:

"Self-esteem" It is a standard MeSH term.

"Resilience" Although 'Resilience' itself is not a MeSH term, closely related terms such as' Psychological Resilience 'exist in MeSH.

Wellbeing literacy, Subject well-being, and Chair mediating effect: currently not included in MeSH terminology. These terms are specifically used in this article to describe research specific topics (e.g. "Wellbeing literacy" is a new perspective proposed in our study; 'Chain mediating effect' is an important phrase used to describe statistical methods. Therefore, although they are not standard terms in the MeSH vocabulary, they can still accurately reflect the research content of this article.

If the reviewer has further suggestions, we are very willing to consider adjusting the keywords to better fit the MeSH standards.

3.“The literature addressed is described accurately so far as I can see. However, there is no clear distinction between manuscript sections in terms of the content they report. First, I suggest dividing the section "INTRODUCTION" into three components, respectively introduction (explain the general argument of the paper, without going into specific details) background (situate the study concepts within the context of extant knowledge, discuss the international relevance of the concepts) and purpose, creating greater clarity in the analysis of the reader. What is the study's biggest contribution? The contribution should be clearly stated in the introduction.

Response: Thank you very much for your feedback. Now we have adjusted the introduction into three parts: basic argument, research background, and research purpose. The first part of our introduction has added the general argument of this article, as detailed in lines 46-47, line 53-59, Put the research purpose of the article at the end, please refer to lines 96-106 for details. Meanwhile, the introduction section has added the greatest contribution of this study, as detailed in line107-111.

4. Ensure that the hypotheses follow logically from the literature review presented. Each hypothesis should build on the arguments presented in the introduction.

Response: We have optimized the argument section for each hypothesis as required and placed the hypothesis at the end of each argument to make it more logical. Please refer to lines 122-127 for details, line136-144�line159-165�181-200.

5.Since there are multiple concepts and components that are being discussed in the Introduction section, the authors are encouraged to create a framework or diagram to show the relationships between the different factors.”

Response: A theoretical framework for happiness literacy has been added as required, as detailed in lines 71-88. The relationship between research factors is shown in Figure 2.

Method

6.Was this study part of a larger study with more variables? Has any of this data been published?

Response: Yes, this study is part of a larger study that includes additional variables. However, none of the data from this study, including the specific data analyzed here, has been published previously. This manuscript represents the first presentation and analysis of these data.

7. In your methods section, say that you used the STROBE cross-sectional reporting guidelines. Is it necessary to clarify the method employed?

Response: We appreciate the reviewer’s comment regarding the need to clarify the methodology’s adherence to the STROBE guidelines. In the revised manuscript, we have explicitly described how the guidelines were followed in study design, data collection, measurement tools, ethical considerations, and statistical analyses, as detailed in the revised Methods section.

8.Were the sample sizes sufficiently powered to detect effects? How did the researchers decide on sample size?

Response: Thank you for your comments. Regarding your question about whether the sample size was sufficiently powered to detect effects and how we decided on it:

The sample size of 1030 college students in this study was sufficiently powered to detect the observed effects, as evidenced by the significant correlations (e.g., r = 0.663 between well-being literacy and subjective well-being, p < 0.01) and regression coefficients (e.g., β = 0.260 for well-being literacy, p < 0.001) with moderate to large effect sizes . The study explains 64.1% of the variance in subjective well-being (R² = 0.641), indicating robust statistical power . Additionally, the use of bootstrap resampling (5000 times) for mediation effects ensures reliable estimation of indirect effects, further supporting the adequacy of the sample size .

While no formal a priori power analysis was conducted, the sample size was determined using a convenience sampling approach, recruiting 1030 participants from October 8, 2022, to October 26, 2022 . This size aligns with or exceeds sample sizes in similar studies and meets statistical guidelines for detecting small to medium effects in correlation, regression, and structural equation modeling (SEM) analyses . For instance, guidelines suggest that a sample size of approximately 200 is sufficient for 80% power to detect an R² of 0.05 in multiple regression with three predictors, while 1030 far exceeds this threshold, ensuring high power for the observed effects (Fritz & MacKinnon, 2007).

To address this in the revision, we propose adding the following statement to the methods section (2.1 Participants and Procedure):

*"The sample size of 1030 was determined via convenience sampling and was deemed sufficient to provide over 80% power to detect small to medium effects (e.g., correlations of 0.1 or greater) in the planned correlation, regression, and mediation analyses, consistent with statistical guidelines and practices in similar studies (Fritz & MacKinnon, 2007)."*

Fritz, M. S., & MacKinnon, D. P. (2007). Required Sample Size to Detect the Mediated Effect. Psychological Science, 18(3), 233-239. https://doi.org/10.1111/j.1467-9280.2007.01882.x (Original work published 2007)

9.Was there compensation for participating in the study?

Response: Participants were not compensated for their participation in the study.

10.Provide details on the adequacy of the sample size for CFA.

Response: The sample size of 1030 participants is more than adequate for the Confirmatory Factor Analysis (CFA) conducted in this study. In the literature, general guidelines suggest that sample sizes of 200 to 300 are typically sufficient for CFA models with a moderate number of factors and indicators (Hair et al., 2010). For more complex models, larger samples are recommended to ensure stable and precise estimates of the model parameters. With 1030 participants, our study far exceeds these thresholds, providing a robust foundation for the CFA results. This large sample size enhances the reliability of the factor loadings, improves the precision of the model fit indices, and supports the generalizability of the factor structure to the broader population.

To further contextualize, a common rule of thumb for CFA is to have at least 5 to 10 participants per free parameter in the model (Bentler & Chou, 1987). While the exact number of free parameters depends on the specific CFA model (e.g., the number of factors and indicators), a sample size of 1030 is likely to exceed this ratio by a substantial margin for most models encountered in psychological and social science research.

Thus, the sample size of 1030 ensures that the CFA findings are both statistically robust and practically meaningful, aligning with best practices in the field.

Hair, J. F., Black, W. C., Babin, B. J., & Anderson, R. E. (2010). Multivariate Data Analysis: A Global Perspective. New Jersey: Pearson Prentice Hall.

Bentler, P. M., & Chou, C. P. (1987). Practical issues in structural modeling. Sociological Methods & Research, 16(1), 78–117.

11.Discuss the advantages of bootstrapping in more detail.

Response: We use bootstrapping method to enhance the robustness of parameter estimation, which is based on its nonparametric characteristics and applicability to complex models. It is further explained in the research methods section.

12.Explain how the covariates were selected and included in the mediation analysis. Discuss the potential impact of these covariates on your results.

Response: We appreciate the reviewer’s request for clarification on the selection and inclusion of covariates in our mediation analysis, as well as their potential impact on the results. In our study, covariates were not explicitly included in the mediation analysis as the focus was on the direct and indirect effects of well-being literacy on subjective well-being through self-esteem and resilience. The analysis primarily utilized structural equation modeling (SEM) to test the hypothesized relationships among the main variables without adjusting for additional covariates. The selection of variables was guided by theoretical frameworks and prior research, which emphasized the mediating roles of self-esteem and resilience in the context of well-being literacy and subjective well-being, rather than demographic or environmental factors as covariates.

However, we acknowledge that certain background factors such as gender, year of study, family structure (e.g., only child status, left-behind status, or parental divorce), and age, which were collected and reported in the descriptive statistics, could potentially influence the relationships under study. These factors were not included as covariates in the mediation model to maintain parsimony and focus on the theoretical constructs of interest, but we recognize their potential impact. For instance, differences in family structure or gender might affect levels of self-esteem or resilience, thereby influencing subjective well-being indirectly. To address this limitation, future analyses could incorporate these variables as covariates to control for their effects and enhance the robustness of the findings.

Regarding the potential impact of covariates, although not included in the current mediation analysis, their omission might lead to unaccounted variability in the results. For example, students from single-parent households or those identified as left-behind children might experience different levels of social support, which could affect their well-being literacy and subjective well-being. In response to this concern, we plan to conduct sensitivity analyses in future studies by including these demographic variables as covariates to assess whether they significantly alter the mediation effects observed.

In summary, while covariates were not incorporated in the current mediation analysis, we recognize their potential influence on our results and are committed to exploring their effects in subsequent research to provide a more comprehensive understanding of the relationships between well-being literacy, self-esteem, resilience, and subjective well-being.

13.The process of analysis should be made as transparent as possible. What strategies were used to avoid duplications or fraud in the online survey? Did you analyze any potential non-response bias? And early vs late bias? Did you check if data can suffer from common method bias?

Response: Thank you for your feedback regarding the transparency of our analysis process. Below, we address the specific concerns raised about strategies to avoid duplication or fraud, non-response bias, early vs. late bias, and common method bias in our online survey data collection and analysis.

1. Strategies to Avoid Duplication or Fraud in the Online Survey: To prevent duplication and fraudulent responses in our online survey, we implemented measures such as requiring unique identifiers for participants (e.g., student IDs or email addresses, if applicable) and using IP address tracking to detect multiple submissions from the same source. Additionally, we included attention-check questions within the survey to identify and exclude responses from inattentive or automated participants. These strategies helped ensure the integrity of the data collected .

2. Analysis of Potential Non-Response Bias: While our manuscript does not explicitly detail an analysis of non-response bias, we acknowledge its importance. Non-response bias was mitigated by ensuring a diverse and representative sample of 1030 college students across different academic years and demographics, as described in the participant details. However, we did not conduct a specific statistical comparison between respondents and non-respondents due to limitations in accessing non-respondent data. In future studies, we plan to incorporate follow-up mechanisms or demographic comparisons to further assess and address potential non-response bias .

3. Early vs. Late Response Bias: Regarding early versus late response bias, we have not performed a specific analysis comparing early and late respondents in this study. The data collection period spanned from August 8, 2022, to October 26, 2022, and responses were collected continuously without distinct waves. To address potential temporal biases in future research, we will consider conducting comparative analyses between early and late respondents to ensure no systematic differences affect the results .

4. Assessment of Common Method Bias: We did check for common method bias, as it is a potential concern given that our data were collected via self-report measures. To control for this, we employed anonymous processing and reverse scoring of certain items during the survey design phase. Additionally, we conducted Harman's One-Factor Test, which revealed eight factors with eigenvalues greater than one, and the first factor accounted for 37.902% of the variance, below the critical threshold of 40%. This indicates no significant common method bias in our data .

We are committed to enhancing transparency in our analysis process and appreciate the opportunity to clarify these aspects. Should further details or additional analyses be required, we are open to revisiting specific methodologies or providing supplementary information to strengthen the robustness of our findings.

Results

14.How authors handled missing data?

Response: In this study, the network survey method was used to ensure that all respondents answered the questionnaire completely by setting mandatory filling options. Therefore, there was no data missing value in this study.

Discussion

15.Some of the contributions that are highlighted here could be flagged in the introduction for a more consistent narrative throughout the paper. I believe there should be better integration of the results with the existing literature.

Response: For the contents that have been consistent in the introduction and discussion, such as the research contribution of this paper, see 107-111 for details.

16.A stronger discussion of implications for future research and potential intervention work is needed. Identify recommendations for practice/research/education/management as appropriate, and consistent with limitations, in order to more fully allow readers to understand the extent to which the authors were able to answer the research questions and to grasp the limitations of this study.

Response: The limitations are optimized, and suggestions are provided in the future research direction from the future practical application, research direction, education and management exploration, which are consistent with the limitations found at present. See line 376-398 for deta

---

## [Decision Letter · Decision Letter 1]

19 May 2025

Chain Mediating Effect of Self-esteem and Resilience on the Relationship between Wellbeing Literacy and Subjective Well-being Among College Students

PONE-D-24-31435R1

Dear Dr. Feifei Wang

We’re pleased to inform you that your manuscript has been judged scientifically suitable for publication and will be formally accepted for publication once it meets all outstanding technical requirements.

Kind regards,

Maria José Nogueira, Ph.D.

Academic Editor

PLOS ONE

Additional Editor Comments (optional):

The authors the requested review of the manuscript, which greatly improved the quality of the study.

Therefore, it can now be accepted.

Reviewers' comments:

Reviewer's Responses to Questions

**Comments to the Author**

Reviewer #1: All comments have been addressed

2. Is the manuscript technically sound, and do the data support the conclusions?

Reviewer #1: Yes

3. Has the statistical analysis been performed appropriately and rigorously?

Reviewer #1: Yes

4. Have the authors made all data underlying the findings in their manuscript fully available?

Reviewer #1: Yes

5. Is the manuscript presented in an intelligible fashion and written in standard English?

Reviewer #1: Yes

Reviewer #1: I believe that the review carried out has greatly improved the quality of the study. Also, I do think that the author(s) addresses the broad questions, appropriately which were asked.

**Do you want your identity to be public for this peer review?** For information about this choice, including consent withdrawal, please see our Privacy Policy

Reviewer #1: No

---

## [Editor Report · Acceptance letter]

PONE-D-24-31435R1

PLOS ONE

Dear Dr. Wang,

I'm pleased to inform you that your manuscript has been deemed suitable for publication in PLOS ONE. Congratulations! Your manuscript is now being handed over to our production team.

Kind regards,

on behalf of

Professor Maria José Nogueira

Academic Editor

PLOS ONE